# Impact of *tps1* Deletion and Overexpression on Terpene Metabolites in *Trichoderma atroviride*

**DOI:** 10.3390/jof10070485

**Published:** 2024-07-14

**Authors:** Xinyue Wang, Wenzhe Li, Shuning Cui, Yuanzheng Wu, Yanli Wei, Jishun Li, Jindong Hu

**Affiliations:** Ecology Institute, Qilu University of Technology (Shandong Academy of Sciences), Jinan 250013, China; wangxinyue9908@163.com (X.W.);

**Keywords:** *Trichoderma atroviride*, terpene synthase, chromatography–mass spectrometry, solid-phase microextraction technique

## Abstract

Terpenoids are structurally diverse natural products that have been widely used in the pharmaceutical, food, and cosmetic industries. Research has shown that fungi produce a variety of terpenoids, yet fungal terpene synthases remain not thoroughly explored. In this study, the *tps1* gene, a crucial component of the terpene synthetic pathway, was isolated from *Trichoderma atroviride* HB20111 through genome mining. The function of this gene in the terpene synthetic pathway was investigated by constructing *tps1*-gene-deletion- and overexpression-engineered strains and evaluating the expression differences in the tps1 gene at the transcript level. HS-SPME-GC-MS analysis revealed significant variations in terpene metabolites among wild-type, *tps1*-deleted (Δ*tps1*), and *tps1*-overexpressed (O*tps1*) strains; for instance, most sesquiterpene volatile organic compounds (VOCs) were notably reduced or absent in the Δ*tps1* strain, while nerolidol, β-acorenol, and guaiene were particularly produced by the O*tps1* strain. However, both the Δ*tps1* and O*tps1* strains produced new terpene metabolites compared to the wild-type, which indicated that the *tps1* gene played an important role in terpene synthesis but was not the only gene involved in *T. atroviride* HB20111. The TPS1 protein encoded by the *tps1* gene could function as a sesquiterpene cyclase through biological information and evolutionary tree analysis. Additionally, fungal inhibition assay and wheat growth promotion assay results suggested that the deletion or overexpression of the *tps1* gene had a minimal impact on fungal inhibitory activity, plant growth promotion, and development, as well as stress response. This implies that these activities of *T. atroviride* HB20111 might result from a combination of multiple metabolites rather than being solely dependent on one specific metabolite. This study offers theoretical guidance for future investigations into the mechanism of terpenoid synthesis and serves as a foundation for related studies on terpenoid metabolic pathways in fungi.

## 1. Introduction

Terpenoids are an important class of volatile organic compounds that not only participate in physiological processes such as plant growth and environmental response but also find extensive applications as raw materials in pharmaceuticals, food, and cosmetics due to their pharmacological activities, including anti-HIV-1 virus, inhibition of cancer cell proliferation, and anti-inflammation [1,2,3]. Currently, the primary approaches for acquiring terpenoids involve plant extraction or chemical synthesis; however, these methods are plagued by a low yield, the limited availability of sustainable resources, and environmental unfriendliness. Consequently, the development and application of terpenoids have been somewhat restricted [4,5]. Therefore, it is crucial to explore novel and sustainable resources for terpenoid biosynthesis to advance the discovery and industrialization of the biotechnological production of valuable terpenoids [6,7].

The *Trichoderma* fungus, found in various terrestrial and marine environments, is a significant producer of terpenoids [6,8]. Since 1948, a total of 253 terpenoids have been isolated from *Trichoderma* species and classified into sesquiterpenes, diterpenes, monoterpenes, and meroterpenoid [9]. Some of these compounds have demonstrated medicinal or agronomic importance [10]. For example, harzianum A (HA), a monoterpene, has demonstrated significant cytotoxicity against human cancer cells [11], and the sesquiterpene 3 isolated from *Trichoderma reesei* PSU-SPSF013 exhibits antifungal activity against *C. neoformans* ATCC90112 [12]. As species of *Trichoderma* with biocontrol potential, terpenoids produced by *Trichoderma atroviride* can exert influence on plant growth and development [13,14]. Up until now, the identified terpenic frameworks derived from *T. atroviride* include harziane [15,16], proharziane [16], wickerol [17], cyclonerane [15,18], and synderane [19]. However, the biosynthetic pathways of these compounds have received relatively limited attention in the *Trichoderma* spp.

Terpene synthase (TPS) plays a pivotal role as a regulatory enzyme in the biosynthetic pathway of terpenoids, particularly in governing the remarkable diversity of terpene compounds [20,21]. In organisms, terpenoids can be synthesized through two distinct pathways, the mevalonate (MVA) pathway and the 2-C-Methyl-D-erythritol-4-phosphate (MEP) pathway [22,23]. The MEP pathway has been demonstrated to be present in prokaryotes, chlorella, and higher plants [24], while the MVA pathway exists mainly in eukaryotes [23]. In general, terpenoids are synthesized from two C_5_ isoprene units, dimethylallyl pyrophosphate (DMAPP) and isopentenyl pyrophosphate (IPP), both of which are catalyzed by terpene synthases to produce a wide variety of open-chain or cyclic compounds through substitution, coupling, cyclization, or rearrangement [25,26].

In this study, through the genome sequencing of HB20111 and the subsequent prediction of potential terpene synthases within the genome, a previously unidentified terpene synthase gene, *tps1*, was discovered and subjected to a bioinformatic analysis. To elucidate the function of this gene and identify the terpenes it produces, we manipulated the transcription level and expression of *tps1* by constructing a gene-deletion strain (Δ*tps1*) and an overexpressing-engineered strain (O*tps1*). Furthermore, headspace solid-phase microextraction (HS-SPME-GC-MS) was employed to identify and analyze the characteristics of terpene metabolites in *Trichoderma atroviride* HB20111 and its mutant strains. This enabled us to investigate the role of this gene in the terpene metabolic pathway and provide theoretical guidance for future research on terpene compound synthesis mechanisms in *Trichoderma*.

## 2. Materials and Methods

### 2.1. Strain, Medium, and Culture Conditions

*Trichoderma atroviride* strain HB20111 (conservation number CGMCC16963) was preserved by the Environmental Microbiology Laboratory of the Institute of Ecology, Shandong Academy of Sciences [27]. The pathogenic fungi *Colletotrichum siamense* (CM9), *Fusarium pseudograminearum* (FP), and *Fusarium oxysporum* f.sp. cucumerinum (FOC) were identified and preserved in the Laboratory of Environmental Microbiology, Institute of Ecology, Shandong Academy of Sciences. All strains were cultivated on potato dextrose agar (PDA) under incubation conditions at 28 °C.

PDA medium supplemented with 1 mol/L sucrose (PDA-S) was used for the regeneration of protoplasts, and PDA with 150 μg/mL hygromycin B (PDA-H) was used to screen the *tps1*-deletion and *tps1*-overexpression strains.

The growth rate of *Trichoderma atroviride* HB20111, *tps1*-deficient strain Δ*tps1*, and *tps1*-overexpressing strain O*tps1* were observed using malt extract agar medium (MEA), cornmeal agar medium (CMA), and potato dextrose agar (PDA).

### 2.2. tps1 Gene Cloning and Bioinformatic Analysis

In the previous stage, fine mapping of the *T. atroviride* HB20111 genome was completed by sequencing (not yet publicly available), and then the terpene synthesis gene cluster was identified in the genome of HB20111 by analysis of antiSMASH version 7.0, with the core gene in the cluster being *tps1*. To clarify the evolution process, we performed a protein BLAST with TPS1 (the predicted amino acid sequence encoded by tps1) on the National Centre for Biotechnology Information (NCBI) website (http://www.ncbi.nlm.nih.gov/, accessed on 19 November 2023.), followed by sequence comparison with ClustalX [5] and MEGA11 [28]. The results showed high homology between the *tps1* gene and the *tps* gene of *T. atroviride* IMI 206040, which, coincidentally, is also a core terpene biosynthesis gene cluster in the *T. atroviride* strain IMI 206040.

To verify the function of the *tps1* gene in *T. atroviride* HB20111, the genomic DNA of *T. atroviride* HB20111 was extracted using the E.Z.N.A.^®^ Fungal DNA kit (Omega Bio-Tek, Norcross, GA, USA). The *tps1* gene was subsequently amplified in a 25 µL PCR reaction using the primers TPSF and TPSR and cloned into the pMD-18T vector for the next step of this study. The sequencing results were identified by Sangyo Biotech (Shanghai) Co. (Shanghai, China) The primers were synthesized by Sangong Biotech (Shanghai) Co. The oligonucleotide sequences are shown in Appendix A.

### 2.3. Generation of tps1-Deletion and -Overexpression Transformants

The restriction endonucleases used in this investigation were obtained from Sangyo Biotech (Shanghai). Plasmids for the overexpression studies and plasmids for the knockout studies were provided by Wuhan Miaoling Biotechnology Co. (Wuhan, Hubei, China).

PCR amplification was performed using 2 × Hieff^®^ PCR Master Mix (With Dye) (Yeasen Biotechnology (Shanghai) Co., Ltd., Shanghai, China) and primers 1615Hyg-SF/1615Hyg-SR, resulting in the amplification of a 2000 bp fragment containing approximately 20 bp homologous sequences upstream and downstream of the terpene synthase *tps1* gene as well as the hygromycin resistance cassette (hyg). Subsequently, the hygromycin resistance cassette was inserted at the +625 locus of the *tps1* gene by digesting plasmid pMD18T-*tps1* with *SacII*, leading to the construction of the plasmid pΔ*tps1* (Appendix A). The homologous recombinant fragment for the transformation was then amplified using TPSF/TPSR as primers and pΔ*tps1* as a template. Finally, 10 μL of the purified PCR product was used to transform HB20111 protoplasts for the deletion of the *tps1* gene.

To construct a *tps1*-overexpression vector, primers 1615F1/1615R6 with *BamHI* and *BstEII* cleavage sites were designed for amplifying the *tps1* gene-form plasmid pMD18T-*tps1*. The fragments were digested with restriction endonucleases *BamHI* and *BstEII*, followed by cloning into plasmid pCAMBIA1303-TrpC-Hygro-gpdA (pO*tps1*). The pO*tps1* was linearized with NotI for the subsequent transformation of HB20111 protoplasts.

### 2.4. Transformation of T. atroviride HB20111

The transformation of *T. atroviride* protoplasts was carried out according to the previously described procedures [29]. *tps1*-deleted and -overexpressed transformants were selected by hygromycin B (150 μg/mL) resistance in the genomic DNAs from the transformants themselves, growing after these selective steps. The selected transformants with *tps1* deletion and overexpression were analyzed by PCR using specific primer pairs. For the Δ*tps1* and O*tps1* mutants, the oligonucleotides TPSF/TPSR and Hyg-648F/Hyg-939R (Appendix A) were used to detect a 292 bp fragment of the *hph* gene, while oligonucleotides TPSF and TPSR (Appendix A) were used to detect a 3600 bp fragment corresponding to the recombinant region. Finally, transformants exhibiting the expected PCR bands were selected for further analysis.

### 2.5. Extraction of RNA and Evaluation of tps1 Gene Expression

The expression of the *tps1* genes was examined using real-time quantitative PCR (qPCR) to observe potential changes in a *tps1*-deleted or -overexpressed mutant. *T. atroviride* HB20111, Δ*tps1*, and O*tps1* were grown in PDA for 4 days. Subsequently, the mycelia was washed, ground in liquid nitrogen, and the RNA was extracted using the HiPure Fungal RNA Mini Kit (Magen Biotechnology (Guangzhou) Co., Ltd., Guangzhou, China). The DNase On Column Kit (Magen) was used to remove any residual DNA. This extracted RNA was reverse-transcribed using a 5× Evo M-MLV RT Master Mix Kit (Accurate Biology, Changsha, China). The products were used as templates for a qPCR analysis with actin [30] used as a control reference gene. All the primers utilized for qPCR in this study are listed in Appendix A. Real-time reactions were conducted with the BIO-RADcfx90 Thermal Cycler (Bio-Rad Laboratories, Hercules, CA, USA). Each 20 μL reaction contained cDNA from 100 ng total RNA and 10 μL SYBRGreen qPCR Master Mix (Accurate Biology, Changsha, China) and 200 μM of each primer. Three replicates of each reaction were performed, and melting curves were generated to confirm the amplification results. The relative difference in expression between samples was determined by calculating the 2^−(∆∆Ct)^ ratio between the CT value of the reference gene and that of the target gene.

### 2.6. Measurement of the Terpene Metabolites by Gas Chromatography–Mass Spectrometry (GC-MS)

The cultivation and GC-MS analysis of the various *Trichoderma* strains were performed following previously described methods [31], with minor adjustments. The fungi were pre-cultured on a PDA medium for 2 days. Subsequently, an agar plug with fungal mycelium was excised from the Petri dish and placed centrally on top of a layer of solidified MEA (5 mL) in a 20 mL headspace (HS) vial. After incubation on the MEA medium for 5 days, the HS vials were hermetically sealed using hole caps and PTFE/silicone septa. For the GC-MS measurements, extraction of volatiles was performed using a 50/30 μm DVB/CARonPDMS fiber for HS extraction, and chromatographic separation was carried on an HP5-MS column (30 m × 0.25 mm × 0.25 μm; Agilent, Waldbronn, Germany).

The automated extraction of volatiles was performed using a CTC Trinity autosampler (Agilent Technologies, Santa Clara, CA, USA). Fungal cultures were incubated at 50 °C with shaking for 15 min. Volatiles from the headspace were extracted for 50 min by exposure to an SPME fiber with 50/30μm DVB/CARonPDMS coating without agitation. The volatiles bound to the fiber were transferred to the GC-MS instrument and, further, on the Model 7890B gas chromatograph (Agilent Technologies, CA, USA) coupled to a Model Pegasus BT mass spectrometer (LECO, Laboratory Equipment Corporation., St. Joseph, MI, USA).

The analytes were desorbed for 5 min in a split/splitless injector (operating in splitless mode at 240 °C) equipped with a headspace glass liner (inner diameter of 1.5 mm, Xiangbo, China) and subsequently separated on DB-wax (30 m × 0.25 mm × 0.25 µm). Helium was used as the carrier gas at a constant flow rate of 1 mL/min. The oven program consisted of 40 °C (hold 5 min), 5 °C/min to 220 °C, and 20 °C/min to 250 °C (hold 2.5 min). For mass spectrometry detection, electron impact ionization (EI) was employed, with an energy level set at 70 eV, source 230 °C, quadrupole 150 °C, full scan scanning mode, and mass range of 20~400. Volatiles emitted by HB20111 were provisionally identified through computer searches guided by the National Institute of Standards and Technology (NIST) Mass Spectral Library 2011 (NIST, Gaithersburg, MD, USA).

### 2.7. Growth Assays of Transformants

Growth of the transformants was assessed on MEA, PDA, and CMA media. Wild-type and transformed strains were cultured on PDA, and a 5 mm hole puncher was used to extract samples from the actively growing edge of the colony, which were then placed in the center of the test media. Colony diameter measurements were taken at 1, 2, 3, 4, and 5 days post inoculation. The mean values and standard deviation for all conditions were calculated using Microsoft Excel 2019.

The colony morphology of the strains was observed after 5 days of incubation. Subsequently, the spores were washed down with 10 mL of saline and filtered through two layers of microscope paper. Conidial suspensions from each strain were collected, and hematocrit plates were repeated to facilitate a comparison of spore production among the strains.

### 2.8. Antagonistic Activity

The inhibition rate was determined following a previously described method [32] with minor modifications. Briefly, the bioactivity of the VOCs from HB20111, Δ*tps1*, and O*tps1* against fungal pathogens was assessed using two inverse face-to-face Petri dishes. The Petri dish sandwich was set up with two 9 cm lidless Petri dishes, and the top or bottom of the dual-culture contained 15 mL of PDA inoculated with a 5 mm diameter covered with pathogen mycelium agar plug or test strain mycelium agar in the center. The two lidless plates were immediately sealed with a double layer of parafilm and incubated until the control group was filled with Petri dishes at 28 °C (a pathogen-inoculated plate was used as the control). The colony diameter was measured in millimeters. Each experiment was conducted with three replicates. The percentage of mycelial growth inhibition was calculated according to the following formula:Inhibition of mycelial growth (%) = (Rc − Rt)/Rc × 100%.

The variable Rc represents the diameter of the control colonies, while Rt denotes the diameter of the treatment colonies.

### 2.9. Analysis of Growth-Related Traits in Wheat

The wheat seeds underwent a 30 s treatment with 75% ethanol, followed by a 3 min treatment with 1% sodium hypochlorite. Subsequently, the sterilized seeds were sown in germination bags and supplied with a nutrient solution. The plates containing each mutant strain were incubated for 3 days. Subsequently, the growth-promoting effects of the mutant strains Δ*tps1* and O*tps1* on wheat were evaluated by measuring root length, plant height, fresh weight, chlorophyll content [33], and soluble sugar content [34] 7 days after germination using the wild-type *T. atroviride* HB20111 strain as a control. 

### 2.10. Statistics

The statistical analyses were conducted using SPSS Statistics 25.0 (IBM Corp., Armonk, NY, USA). An analysis of variance (ANOVA) was performed, followed by Tukey’s and Duncan’s tests to identify significant differences among the samples (*p* < 0.05).

## 3. Results

### 3.1. Phylogenetic and Sequence Comparison

Based on the BLASTP analysis, TPS1 (accession number: PP713120) demonstrates a high sequence identity of 92.57% with the sesquiterpene synthase from *Trichoderma atroviride* IMI 206040, indicating a closely related homology. Additionally, the multiple sequence alignment indicates the presence of conserved motifs that are characteristic of typical terpene synthases (Figure 1). The two highly conserved aspartate-rich motifs DDXXD (residues 90–94) and NSE/DTE (residues 237–245), which are commonly found in sesquiterpene synthases, are depicted in Figure 1, along with the universally conserved RXR motif (residues 191–193). The DDXXD and NSE/DTE motifs have been documented to flank the entrance of the active site [35]. They are involved in coordinating a trinuclear magnesium cluster, with DDXXD coordinating two magnesium ions and NSE/DTE coordinating one magnesium ion [25]. These two regions indicate the protein’s capability to catalyze the cyclization of farnesyl diphosphate. The phylogenetic analysis of TPS1’s deduced amino acid sequence reveals a significant homology with *T. atroviride* terpene synthase (Figure 2). Taken together, our findings suggest that *tps1* is a plausible candidate gene responsible for encoding the sesquiterpene synthase in *T. atroviride*.

### 3.2. Generation of tps1 Deletion and Overexpression Transformants

To generate the *tps1*-deletion mutant, the selective marker gene *hph* was located in the middle of the *tps1* gene through protoplast transformation. The stability testing of 24 transformants was conducted on hygromycin resistance plates. The initial PCR screening, using primers TPSF/TPSR (Appendix A), provided preliminary evidence indicating that the *tps1* gene deletion cassette had integrated into the *T. atroviride* HB20111 genome through homologous recombination in two transformants (Figure 3a). A subsequent qRT-PCR analysis confirmed the absence of *tps1* transcript expression in the deletion mutant Δ*tps1*-16, indicating the successful replacement of the *tps1*-coding region through homologous recombination (Figure 3b).

The *tps1*-gene-overexpression strains were also obtained through the transformation of HB20111 protoplasts with linearized plasmids pO*tps1* using *NotI*. Fourteen transformants resistant to hygromycin (150 µg/mL) were screened using primers Hyg-648F/Hyg-939R (Sangyo Biotech (Shanghai) Co., Shanghai, China). All transformants tested positive for the *hph* gene. The *tps1* expression cassette from plasmid pO*tps1* may have been randomly integrated into the genome of *T. atroviride* HB20111 in these overexpressing transformants. Transformant O*tps1*-15 was selected for further characterization and analyzed by qRT-PCR. Expression of the *tps1* gene in O*tps1*-15 was elevated 22-fold in comparison with WT (Figure 3b). Finally, two deletion mutants and overexpression strains were obtained and designated as Δ*tps1* and O*tps1* for subsequent experiments.

### 3.3. GC-MS Analysis

A GC-MS analysis was performed to determine the function of the *tps1* gene. In this analysis, volatiles extracted from the wild-type, deletion strains, and overexpression strains were examined using the HS-SPME method, and three biological replicates of each strain were performed to ensure the accuracy of the experimental results. Under conditions where the wild-type was used as a control group, the HS-SPME-GCMS analyses showed that the deletion or overexpression of the *tps1* gene did not affect the production of dimethyl sulfide, furan, 2-ethenyl-, n-octane, and other non-terpene VOCs (Electronic Appendix A), but the deletion or overexpression of the *tps1* gene affected the production of terpene VOCs.

Two major classes of terpene volatile compounds were produced in the wild-type strain: eight monoterpenes and two sesquiterpenes (Appendix A), with the major compounds being 4-methyl-2(5H)-furanone, β-phellandrene, geranylacetone, α-phellandrene, α-angelica lactone, and acorenone b, which comprised 88.61% of the total terpene volatiles (electronic Appendix A). The presence of (5e,9e)-farnesyl acetone and beta-cedrene was not detected in the deletion mutant compared to the control WT (Appendix A). Interestingly, both the deletion mutant and the overexpression mutant produced new sesquiterpene compounds, trans-nerolidol and nerolidol, respectively, at RT 31.5329 (Appendix A). This suggests that the *tps1* gene may be involved in terpenoid synthesis. Additionally, a single volatile compound, 1,3,6,10-cyclotetradecatetraene, 3,7,11-trimethyl-14-(1-methylethyl)-, (s-(e,z,e,e))-, exhibited a unique presence in the *tps1*-gene-deletion or -overexpression strains, despite its low relative abundance. Concurrently, we observed that the terpene spectra of overexpressed inverters exhibited an increase in content for only four out of the nine sesquiterpenes when compared with those of the wild-type, indicating that the overexpression of the *tps1* gene did not result in a complete enhancement in volatile organic compound production. Conversely, monoterpene species were reduced in all the overexpressed strains, with only one monoterpene (2, 3-dihydro-3, 5-dihydroxy-6-methyl-4h-pyrano-4-one) being comparable to the wild-type (Appendix A). Furthermore, this study also revealed that four sesquiterpenes (nerolidol, Epizonarene, β-acorenol, guaiene) showed a significant increase in abundance of the O*tps1* strain with *tps1* gene overexpression compared to Δ*tps1* gene deletion (Appendix A).

### 3.4. Preliminary Phenotypic Analysis of Deletion and Overexpression Mutants

To assess potential differences in growth among the deletion or overexpression strain (Δ*tps1* and O*tps1*) and the wild-type, strains were cultivated on MEA, PDA, and CMA (Figure 4). Both mutants exhibited at a significantly slower rate than the wild-type on MEA media, while their growth rates on PDA and CMA did not show significant variations compared to the wild-type (Figure 4). Previous studies have suggested that a sluggish growth rate in an organism may be attributed to mutations in genetic factors [36]. This finding also validates the successful construction of engineered strains with *tps1* deletion or overexpression, demonstrating their impact on biological growth.

In addition, to verify whether the *tps1* gene is a key gene affecting the synthesis of conidial pigments in *T. atroviride*, we also observed and analyzed the colony characteristics of the wild-type strain of *T. atroviride* and its mutant strains cultured on PDA plates for 5 d (Figure 5A). The results showed that there was no significant difference between the Δ*tps1* deletion strain and the O*tps1* overexpression strain with denser colonies and more green pigment produced by the conidia, using the wild-type as a control (Figure 5A). The results suggest that the overexpression of *tps1* may accelerate the synthesis of pigments in the conidia of *T. atroviride*. The molecular spore production of the colonies was determined under the same culture conditions, and it was found that there was no significant difference between the Δ*tps1* mutant strain and the WT control strain (Figure 6), suggesting that the deletion of the *tps1* gene did not affect conidial production, whereas the overexpression of the *tps1* gene decreased conidial production. However, the role of the tps1 gene in promoting *T. atroviride* HB20111 growth remains unknown. However, there is evidence that terpenes may affect fungal growth [37]. Therefore, it is hypothesized that the *tps1* gene may have the potential to regulate the composition and amount of terpenes produced by *T. atroviride* HB20111, thereby affecting mycelial growth.

### 3.5. Preliminary Phenotypic Analysis of Deletion and Overexpression Mutants

We also evaluated the effect of *tps1* deletion or overexpression on the fungicidal potential of *T. atroviride* against pathogenic fungi. The results showed that the wild-type progenitor strains and the Δ*tps1* and O*tps1* mutants had inhibitory effects on all the tested pathogens (Figure 7, Table 1). Notably, Δ*tps1* exhibited higher inhibitory activity than the other two Trichoderma species on CM9 and FP, while O*tps1* showed lower inhibitory activity. In terms of FOC inhibition, O*tps1* demonstrated superior antifungal efficacy. These findings are in line with previous research on the antifungal properties of *Trichoderma viride* [38].

### 3.6. Growth-Promoting Effect

To investigate the growth-promotion effects of wild strain HB20111 (WT) and its *tps1*-deletion mutant Δ*tps1*, as well as the *tps1*-overexpression mutant O*tps1* on the plant, a pot experiment was conducted using wheat (JiMai 22) to discover the role of terpenoids in *Trichoderma*–plant root interaction. The experimental results indicated that the deletion or overexpression of the gene did not significantly enhance wheat growth when compared to the control group (WT). However, the content of chlorophyll b, total chlorophyll, and soluble sugars in the leaves exhibited a significant increase (Figure 8).

## 4. Discussion

In the present study, we provide evidence that the *tps1* gene in *T. atroviride* HB20111 is associated with the biosynthesis of sesquiterpenoids. Our results also suggest that the *tps1* gene directly or indirectly affects the expression of genes involved in other secondary metabolite biosynthetic pathways.

The deletion of *tps1* almost eliminated the production of terpinolene, farnesylacetone, β-cedrene, β-phellandrene, α-angelica lactone, and acorenone b in *T. atroviride* (Appendix A). This finding is consistent with previous studies showing that the deletion of terpene synthase reduces terpene production [39,40]. Previous studies have elucidated various functions of these terpenes. For instance, terpinolene has demonstrated inhibitory effects on acetylcholinesterase (AChE) expression and the suppression of gray mold mycelium growth [41,42,43]; the application of farnesyl acetone results in a significant reduction in the individual development observed in the little cabbage moths (*P. xylostella*) [44]; β-cedrene possesses natural diuretic and anti-inflammatory properties, which may aid in cellulite improvement [45]; β-phellandrene acts as an insect sex pheromone, disrupting insect behavior [46]; α-angelica lactone inhibits mouse tumor growth induced by benzopyrene in mice [47]; and pure acorenone b exhibits inhibitory activity against both acetylcholinesterase (AChE) and butyrylcholinesterase (ChE), showing potential for the treatment of Alzheimer’s disease [48]. These experimental findings provide valuable insights into the biological functions of terpenoids within *T. atroviride* HB20111.

Moreover, deletion of the *tps1* gene resulted in significant changes in terpenoid species and content, which significantly affected the antifungal activity of *T. atroviride* HB20111 against *Colletotrichum siamense*, *Fusarium pseudograminearum*, and *Fusarium oxysporum* f.sp. cucumerinum. This is in line with previous findings that plants such as cotton (*Gossypium* species) and Camphor cinnamon may produce fewer antimicrobial terpenoids in the absence of certain terpene synthase genes, thus weakening their defense mechanisms against fungal pathogens [49,50,51]. The main reason for this is that, when terpene synthase genes are disrupted or missing, this leads to a reduction in the biomass of specific terpenoids or the production of different and sometimes less-effective compounds. In addition, the terpenes encoded by the *tps1* gene are believed to play a crucial role in promoting plant growth through enhancing root system development. Further scientific investigations are required to clarify the underlying molecular mechanism. Furthermore, we hypothesize that the antifungal efficacy of *T. atroviride*’s VOCs may be attributed to the biosynthesis of two or more structurally distinct secondary metabolites. The cessation of one metabolite’s production does not compromise antifungal activity, as the synthesis of other metabolites compensates for the deficiency. 

This study also conducted a comparative analysis of terpene metabolites among wild-type, deletion strains, and overexpression strains, revealing significant variation in the terpene profiles across the different strains. The *tps1*-deleted mutants exhibited a marked reduction in or absence of most sesquiterpene VOCs, confirming the potential involvement of the *tps1* gene in sesquiterpene compound biosynthesis. Additionally, both the deletion and overexpression strains yielded novel terpene metabolites; however, not all volatile organic compounds were detected in both strains, particularly nerolidol, β-acorenol, and guaiene, which were exclusively produced by the O*tps1* strain. A previous study by Liu et al. reported that the terpene cyclase *tarA* is involved in the production of the sesquiterpene compound tricinoloniol acids (TRAs) in *Trichoderma hypoxylon* [52]. The BLAST analysis showed that TPS1 shared a sequence similarity of 76.88% with QNC71690.1 from *T. hypoxylon* at the amino acid level. Further phylogenetic analysis also revealed their homology (Figure 2). Considering the TRA synthesis pathway [52], it is hypothesized that TPS*1* may function similarly to TraA proteins as a terpene cyclase involved in sesquiterpenoid synthesis. The bioinformatic analysis indicated that the TPS1 protein belongs to the terpene cyclase-like 2 subgroup and contains an encoded terpene cyclase, three metal sites, and a hydrogen bond bound to the pyrophosphate anion. Referring to previous studies, a putative biosynthetic pathway for the formation of TPS1 was proposed: TPS1 acts as a terpene cyclase with a building block of farnesyl pyrophosphate (FPP) to produce beta-cedrene. It is worth noting that a single terpene synthase may generate multiple products; for instance, the γ-humulene synthase from North American fir has the capacity to produce 52 different sesquiterpenes, possibly due to distinct binding orientations within the substrate-binding DDXXD structural domain [53]. Therefore, if a reaction permits the formation of multiple intermediates, it is more likely that some side products requiring fewer steps or shorter reaction times than the main product will be formed [54].

As far as we know, the biological functions of fungal sesquiterpenes in relation to themselves or fungal–plant interactions remains incomplete. Current research primarily focuses on sesquiterpenes in plants, which produce terpenoids which function as phytotoxic and repellent compounds against phytophagous insects. For instance, Melia azedarach releases neemin, a furan triterpenoid, to deter insects such as *Pieris occidentalis* and *Meucania separata* [55,56]. Fungal sesquiterpenes have similar functions but have been less extensively studied [57].

## 5. Conclusions

In conclusion, the findings of this study suggest that *tps1* plays a regulatory role in the expression of terpene synthases, which is crucial for understanding the biosynthetic mechanism of sesquiterpene compounds. Additionally, it appears that *tps1* genes also modulate the expression of the pyruvate metabolic pathway genes. Furthermore, while the wild-type *T. atroviride* HB20111 strain produces sesquiterpene compounds with antifungal activity, their production may not be essential for this activity. This implies that the antifungal activity of the fungus may result from a combination of multiple metabolites or other factors working together rather than being solely dependent on one specific metabolite. Moreover, these experimental findings have the potential to serve as a foundation for genetic modification in *T. atroviride* HB20111. Therefore, it is suggested that *tps1* in HB20111 has broader effects beyond a direct involvement in sesquiterpene compound production or the synthesis of metabolic precursors, potentially influencing gene expression related to plant growth and development as well as the stress response; however, further research is needed to confirm these hypotheses.

## Figures and Tables

**Figure 1 jof-10-00485-f001:**
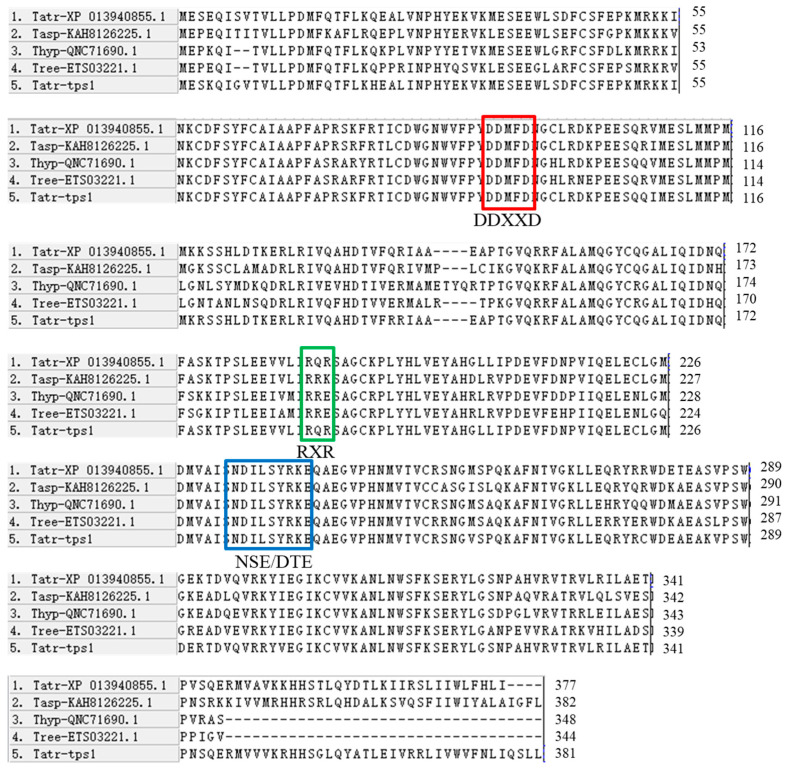
Multiple-sequence alignment of TPS1. The deduced TPS1 amino acid sequence of TPS1 and terpene synthase of other species (*T. atroviride* IMI 206040, *T. asperelloide*, *T. hypoxylon*, and *T. reesei* RUT C-30). The color box corresponds to the region of each conserved domain (DDXXD, NSE/DTE, and RXR).

**Figure 2 jof-10-00485-f002:**
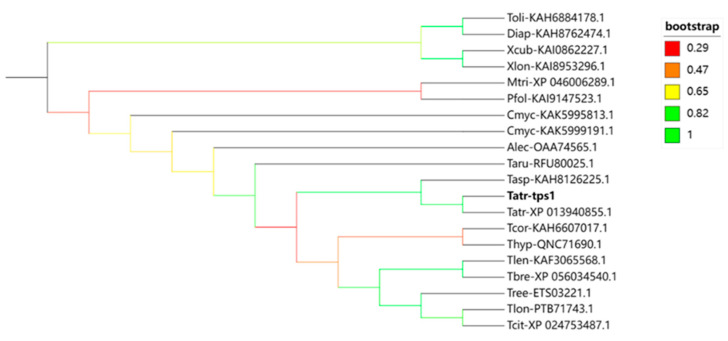
Phylogenetic analysis of TPS1. Phylogenetic tree of the protein sequences of 20 genes encoding terpene synthase from 19 fungal species. The species are *T. atroviride* (Tatr), *T. breve* (Tbre), *T. lentiforme* (Tlen), *T. citrinoviride* (Tcit), *T. cornu-damae* (Tcor), *T. longibrachiatum* (Tlon), *T. reesei* (Tree), *T. hypoxylon* (Thyp), *T. asperelloides* (Tasp), *T. arundinaceum* (Taru), *Thelonectria olida* (Toli), *Diaporthaceae* sp. (Diap), *Xylaria cubensis* (Xcub), *Xylaria longipes* (Xlon), *Microdochium trichocladiopsis* (Mtri), *Paramyrothecium foliicola* (Pfol), *Cladobotryum mycophilum* (Cmyc), *Akanthomyces lecanii* (Alec), and *Trichoderma cornu-damae* (Tcor). The numbers following the abbreviated names correspond to NCBI numbers. The terpene synthase from *T. atroviride* HB20111 was labeled Tatr-*tps1*.

**Figure 3 jof-10-00485-f003:**
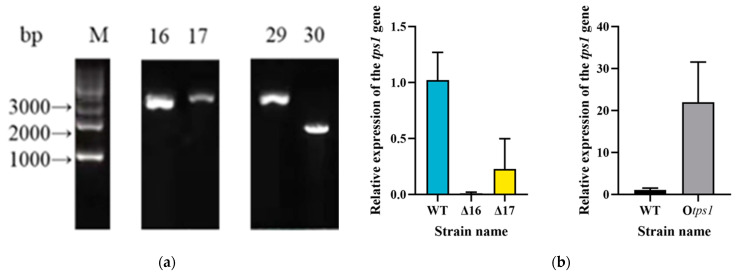
qRT-PCR analysis and confirmation of *tps1* transformants. (**a**) The principle of homologous recombination was utilized to delete the terpene synthase gene *tps1* by inserting the screening marker *hph* gene. Following amplification with oligonucleotide TPSF/TPSR, a 1645 bp fragment of the *tps1* gene was produced in the wild-type strain, while a 3600 bp recombinant fragment was generated in the deletion strain (M—GL 5000 Maker; 16, 17—amplification of recombinant fragments of picked transformants; 29—pMD18T-Δ*tps1*-*hyg* plasmid; 30—wild-type strain HB20111 genome). (**b**) qRT-PCR analysis of *T. atroviride* wild-type (HB20111) strain, *tps1*-deletion mutants (16, 17), and *tps1*-overexpression mutants.

**Figure 4 jof-10-00485-f004:**
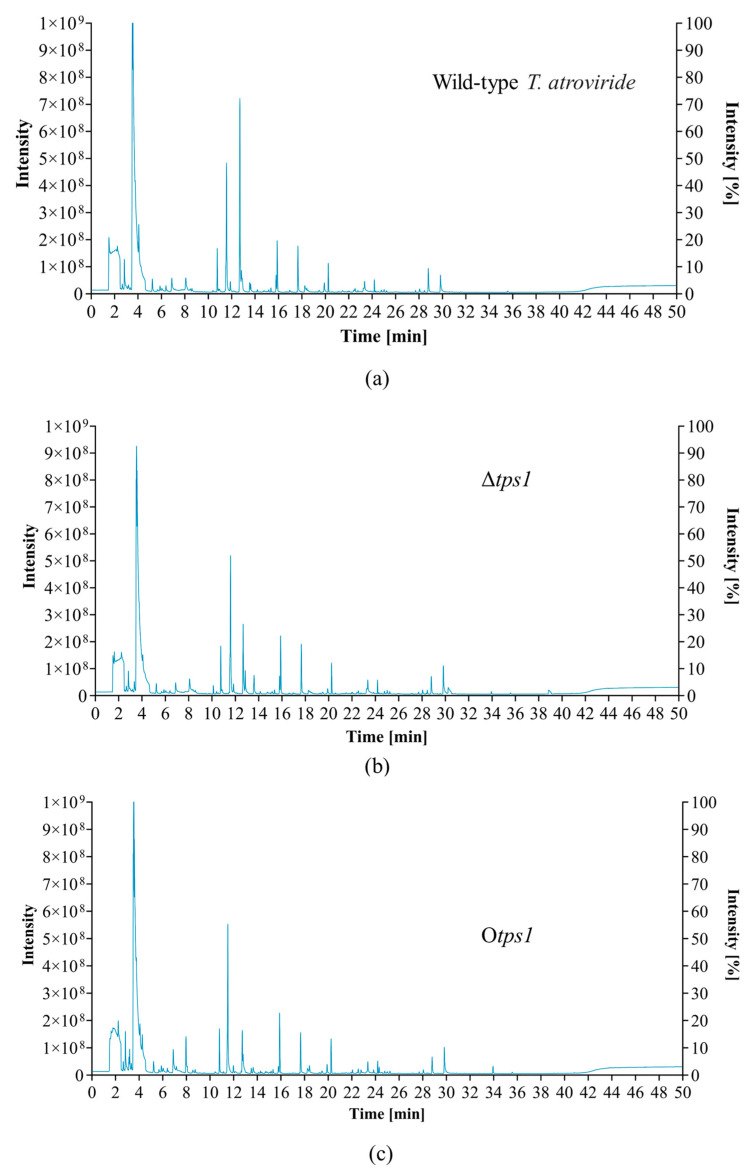
GC-MS chromatograms of wild-type 20111 (**a**), *tps1*-deleted mutant (**b**), and *tps1*-overexpressed mutant (**c**).

**Figure 5 jof-10-00485-f005:**
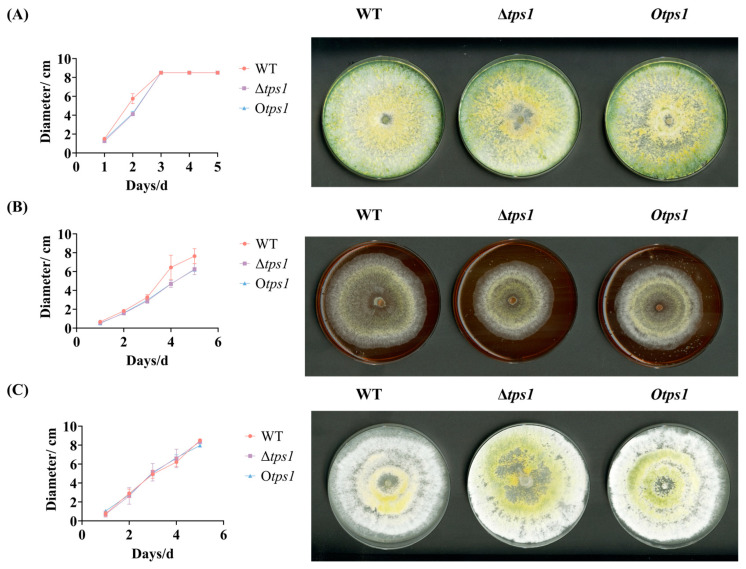
Growth of terpenoid-synthase-deletion and -overexpression transformants. HB20111, Δ*tps1*, and O*tps1* were individually cultured on (**A**) potato dextrose agar (PDA), (**B**) malt extract agar (MEA), and (**C**) cornmeal agar medium (CMA). The cultures were incubated for five days, followed by measurement.

**Figure 6 jof-10-00485-f006:**
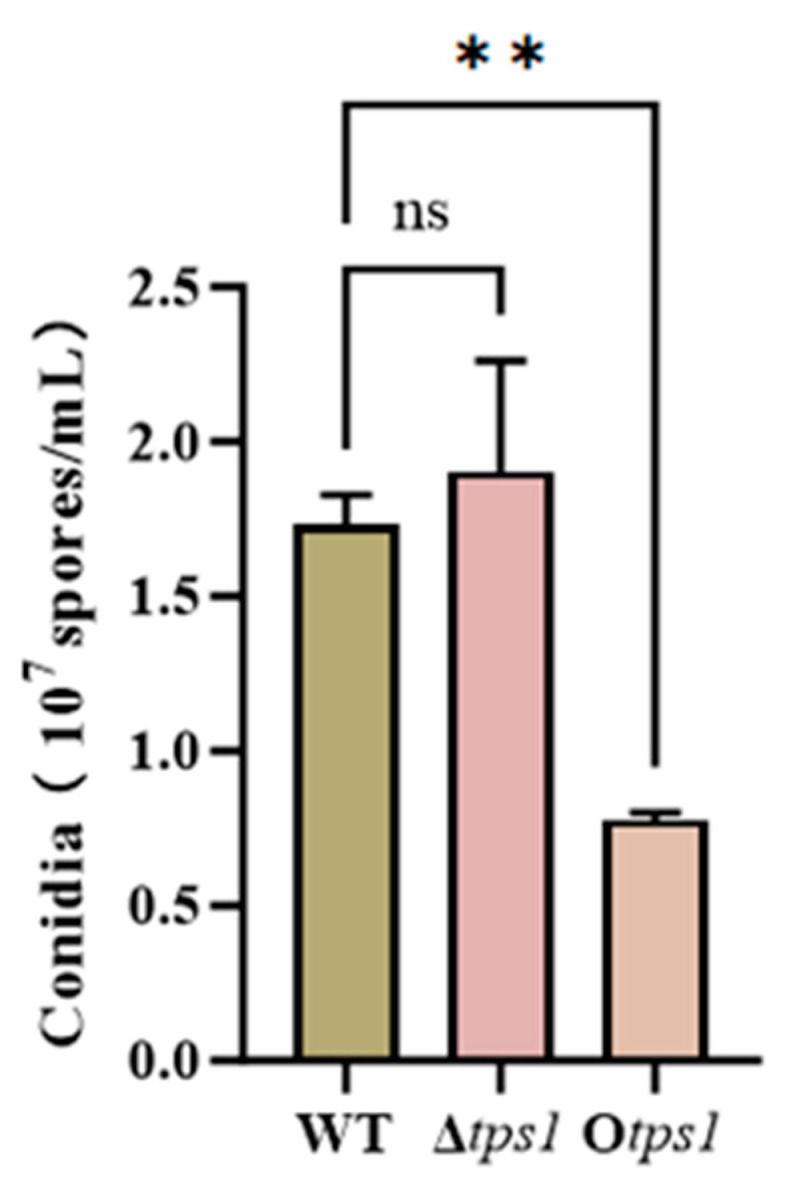
Conidia production of WT, Δ*tps1*, and O*tps1* strains. The value bars with ns are not significant, asterisk denotes statistically significant differences (** *p* < 0.01).

**Figure 7 jof-10-00485-f007:**
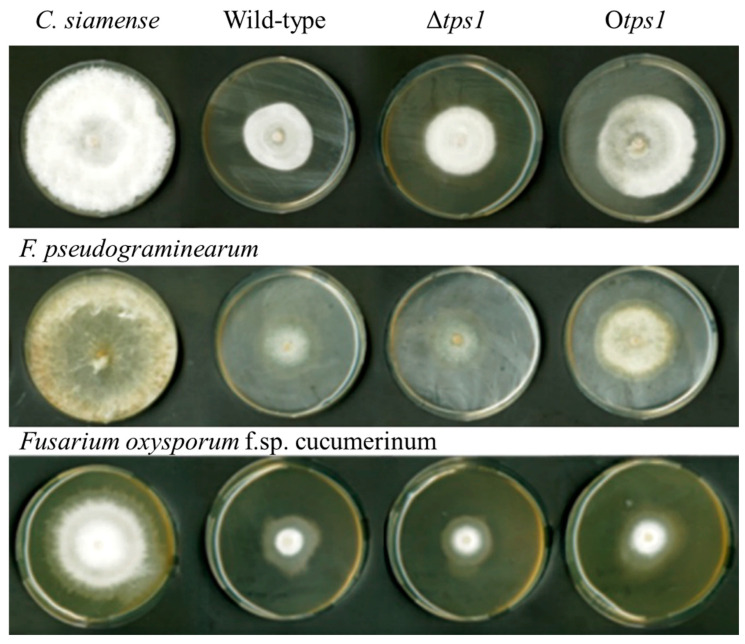
Antifungal activity of wild-type *T. atroviride* HB20111and its mutant strains.

**Figure 8 jof-10-00485-f008:**
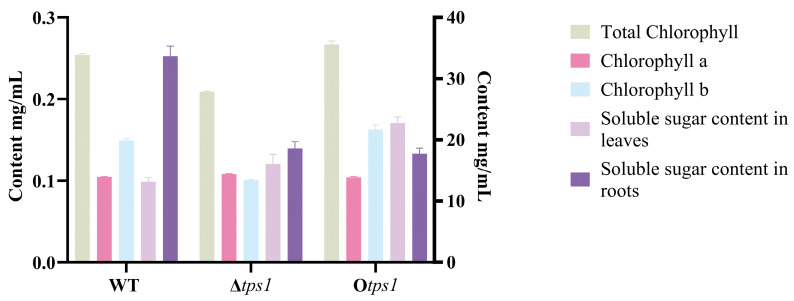
Effects of 20111 and its mutant on chlorophyll and soluble sugar content in wheat under 28 °C light condition.

**Table 1 jof-10-00485-t001:** Inhibition of mycelial growth (RI) caused by HB20111, Δ*tps1*, and O*tps1* against *Colletotrichum siamense* (CM9), *Fusarium pseudograminearum* (FP), and *Fusarium oxysporum* f.sp. cucumerinum (FOC), grown on PDA.

	*C. siamense*	*F. pseudograminearum*	*Fusarium oxysporum* f.sp. cucumerinum
WT	34 ± 0.8185 b *	48.367 ± 0.7506 c	36.133 ± 2.7538 b
Δ*tps1*	50.433 ± 0.611 a	55.567 ± 1.35 a	34.167 ± 4.7543 b
O*tps1*	27.8 ± 2.0811 c	51 ± 0.2 b	43.667 ± 2.3029 a

* ANOVA and Tukey’s test. For each column, values followed by different superscript letters are significantly different (*p* < 0.05).

## Data Availability

All newly generated sequences were deposited in GenBank (https://www.ncbi.nlm.nih.gov/genbank/ (accessed on 1 May 2024)).

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
