# Peer review of "Impact of tps1 Deletion and Overexpression on Terpene Metabolites in Trichoderma atroviride"

_jof, 2024, doi:10.3390/jof10070485_

Round 1

Reviewer 1 Report

In this manuscript, Wang and colleagues isolate the tps1 gene, encoding a terpene synthase, from the strain HB20111 of the fungus Trichoderma atroviride. The authors construct isolates lacking or overexpressing tps1 and assess their effect on terpene metabolites among others. As the authors acknowledge in one section of the article “Preliminary phenotypic analysis of deletion and…” this seems to be a study in progress that requires further experimentation. Additionally, the writing should be improved as many clarifications and language editing are needed. Please find me comments below.

1.        The most interesting piece of data is the antifungal effect shown in Fig. 6 in which altering tps1 levels renders different outcomes. The authors link this effect to change in terpene composition, but no direct evidence is provided.

2.        Was the HB20111 strain genome sequence already available or the authors sequenced it for this project?

3.        The tps1 gene from other T. atroviride strain (IMI 206040) was known. What is the rationale for isolating tps1 in HB20111 strain. 

4.        Following from above. What is the difference in terms of phenotypes and terpene composition when compared mutants or overexpression from both strains?

5.        It is not clear what accession TPS1(ACCESSION PP713120) means? Is it from T. atroviride HB20111 genome? Please clarify that.

1.        Materials and Methods: Subheadings cannot be repeated. Otherwise, fuse the paragraphs.

2.        Define GC-MS, MEA, PDA, CMA (line 309 states CMD) and FOC

3.        Font in figures 3b, 4 and 5 is too small.

Language editing:

_ Inappropriate use of uppercase. Examples as follows:

Line 22 and 240: “Fungal”

Line 40: “Consequently”

Line 249: “Transformants”

_ Grammar:

Line 345: “Effect of 20111 and it’s mutant”

_ Punctuation:

Line 257: “NotI..”

_ Gene Nomenclature. Please be consistent. 

Line 249: “tps1” no italics

Line 251: “tps

Line 411: “tp1

Reviewer 2 Report

The manuscript entitled «Impact of tps1 Deletion and Overexpression on Terpene Metabolites in Trichoderma atroviride» is devoted to the study of the role of the tps1 gene coding terpene synthase as well identification terpene metabolites in Trichoderma atroviride.

Although the manuscript is carefully written and the presented experiments seem to be well conducted, however, there are few questions to the authors.

7. Unfortunately, I didn't have access to the supplementary Table S1 and Table S2, making a final review impossible. I can only see Figure S1. There was probably some kind of failure in the process of downloading of supplementary files.

8. There are a lot of small typos in the text, you need to check them carefully. I tried to mark them in the pdf file in correction mode. Pay attention to the comments.

1. Given the dependence of secondary compound production on random external factors, and the dependence of the resulting data on the sample preparation process, the question arises: how many biological (from the different strain samples) and analytical replicates were there in the GC-MS analysis of the volatile triterpenoids?

2. Figure 1 depicts an alignment of TPS1 from 5 fungal species, among them 2 from T. atroviride. In my opinion, it would be logical to place these two sequences side by side, under numbers 1 and 2. Why are the grey row containing the headings and asterisks that mark the conservative amino acids of the alignment shown only in the first and third rows? And RXR has moved to the right of the green frame.

3. Figure 3 (b) would be good to enlarge a bit and improve the resolution.

4. P. 274-279  Please rewrite the paragraph more clearly. In this analysis, material extracted from the different strain samples was examined using the HS-SPME method to ensure the accuracy of the experimental results. What a strain? Wt or transformants? That results?

The GC-MS analysis yielded three chromatograms with similar patterns, indicating that the deletion or overexpression of the tps1 gene did not impact the production of other volatile organic compounds . Others relative to what?

5. P. 287, 347. This confirmed the direct involvement of tps1 in sesquiterpene volatile biosynthesis. Why direct?

6. Figure 5. Please mark on the panel with cultures photo or in the figure footnote where are wt, Δtps1, and Otps1 culture photographs. Is it just me or did the deletion affect the morphology of the colonies, specifically the formation of a pigmented crust? (If I've guessed and Δtps1 strains photos are in the middle). Maybe the TPS1 is somehow related to the mycelium differentiation process?

Round 2

Reviewer 1 Report

The current version is considerably improved.

Prior publication, please amend the following items:

_ Subheading 2.7 and 2.8 are still the same.

_ Figures 4 and 5. Graphs font is still too small.

_ Figure 8 legend: Please state "Effect of indicated strains on chlorophyll and ..." Please remove "bacteria".

Reviewer 2 Report

The authors have responded to all comments and have greatly improved their manuscript.

I think, the manuscript can be accepted after minor corrections:

Line 23. impact on Fungal inhib… - it doesn't need a capital letter.

Lines 130, 143, 219.  - no space in T.atroviride.  T._atroviride

Line 343. pathogens(Figure 7, Table 1). - no space pathogens_(Figure 7, Table 1).

Figure 7,  Table S1. Fusarium oxysporum Cucumerinum owen  - misspelled species name. The full correct name of this fungus is:  Fusarium oxysporum f.sp. cucumerinum J.H. Owen 1956 

But as you don't cite authors for other species (F. pseudograminearum, C. siamense), you can write without an author: Fusarium oxysporum f.sp. cucumerinum  Note that f.sp. should not be italicized. It's important.

Figure 7. AntiFungal - it doesn't need a capital letter. Antifungal

Table S2. It would be more convenient to express Relative Abundances of terpenes in  nx106 or in persentages.
